# The outcomes of three different techniques of coronary artery bypass grafting: On-pump arrested heart, on-pump beating heart, and off-pump

**Amarit Phothikun**[1,2,3], **Weerachai Nawarawong**[1], **Apichat Tantraworasin**[2,3,4], **Phichayut Phinyo**[2,5,6], **Thitipong Tepsuwan**[1,3] *

1 Faculty of Medicine, Department of Surgery, Cardiovascular and Thoracic Surgery Unit, Chiang Mai University, Chiang Mai, Thailand, 2 Faculty of Medicine, Clinical Epidemiology and Clinical Statistic Center, Chiang Mai University, Chiang Mai, Thailand, 3 Clinical Surgical Research Center, Chiang Mai University, Chiang Mai, Thailand, 4 Faculty of Medicine, Department of Surgery, General Thoracic Surgery Unit, Chiang Mai University, Chiang Mai, Thailand, 5 Faculty of Medicine, Department of Family Medicine, Chiang Mai University, Chiang Mai, Thailand, 6 Musculoskeletal Science and Translational Research (MSTR), Chiang Mai University, Chiang Mai, Thailand

☯ These authors contributed equally to this work.
* tepsuwanthitipong@gmail.com

## Abstract

### Objective

Conventional coronary artery bypass grafting (CABG) or on-pump arrested heart CABG (ONCAB) is a standard and simple technique. However, adverse effects can occur due to the use of aortic cross-clamp and cardiopulmonary bypass. Performing off-pump CABG (OPCAB) aims to avoid these adverse effects but may result in incomplete revascularization. On-pump beating heart CABG (ONBHCAB) combines the benefits of both ONCAB and OPCAB. This study focuses on comparing the short- and long-term outcomes of different CABG techniques.

### Method

Retrospective observational cohort included 2,028 patients who underwent ONCAB, OPCAB, and ONBHCAB. The short-term outcomes including postoperative ischemic injury, hemodynamic functions, and adverse events were compared. The long-term outcomes were overall survival and the occurrence of major adverse cardiovascular events (MACE). Propensity score matching ensured comparability among the three patient groups.

### Results

After matching, there were no differences in baseline characteristics. Regarding ischemic injury, OPCAB showed the lowest peak cardiac enzyme levels (all $p \leq 0.001$). There were no statistically significant differences in the change of hemodynamic function (cardiac index) between the three groups (p = 0.158). Ten-year survival for OPCAB, ONBHCAB, and ONCAB were 80.5%, 75.9%, and 73.7%, respectively. OPCAB was associated with a significant reduction in mortality risk and MACE when compared to others (Mortality HR = 0.33, p = 0.001, MACE HR = 0.52, p = 0.004).

**Data Availability Statement:** The data underlying the results presented in the study are available from Phothikun, Amarit (2023): Full data set for

PLOSONE.xlsx. figshare. Dataset. https://doi.org/10.6084/m9.figshare.22723688.v1.

**Funding:** For the financial disclosure, this work was supported by the Faculty of Medicine, Chiang Mai University, Thailand. The grant number for funding support was no.185/2563. This research did not receive any specific grant from funding agencies in the commercial.

**Competing interests:** The authors have declared that no competing interests exist.

## Conclusion

OPCAB implementation resulted in a lower occurrence of postoperative ischemic injury than ONCAB and ONBHCAB. No differences in postoperative hemodynamic function in all three techniques were observed. OPCAB respectively were preferable techniques beneficial for long-term outcomes.

## 1. Introduction

Coronary artery bypass grafting (CABG) can be performed using three different techniques. The conventional technique, known as on-pump arrested heart CABG (ONCAB), is a widely used and effective procedure [1]. However, this technique still uses cardiopulmonary bypass (CPB), aortic cross-clamp, and a cardioplegic solution to arrest the heart, which can compromise coronary perfusion and cause ischemic injury to the myocardium during anastomosis [2]. CPB has numerous adverse effects, including increased systemic inflammatory responses (SIRs), increased total body fluid, coagulopathy, platelet dysfunction, renal insufficiency, all of which increase postoperative morbidity and mortality [3].

Off-pump CABG (OPCAB) is a technique that aims to avoid adverse effects from both CPB and aortic cross-clamp use [3]. However, OPCAB has some limitations, including the need for specific equipment and experienced surgeons. Moreover, manipulating the heart during the operation may disturb hemodynamics, and performing bypass grafting while the heart is still beating can lead to incomplete revascularization [2].

On-pump beating heart CABG (ONBHCAB) is another technique that avoids the effects resulting from aortic cross-clamp use and cardioplegic arrest. The principle behind ONBHCAB is to ensure continuous coronary perfusion to protect against myocardium injury [4]. This technique uses CPB to provide circulatory support and maintain intra-operative hemodynamic stability. Good exposure of the coronary target can be achieved via heart manipulation without concern of hemodynamic deterioration, contributing to the complete revascularization [5].

The most appropriate CABG technique remains controversial. Prior to this study, several studies had compared the outcomes of OPCAB vs ONCAB [1–3,6], and ONCAB vs ONBHCAB [7–9]. A smaller number of studies compared OPCAB vs ONBHCAB [5,10]. In all of these studies, the main focus was on comparing the use of CPB vs none-CPB, aortic cross-clamp vs no-clamp, and the resulting postoperative complication. The issue of the revascularized technique and the potential for incomplete revascularization was also discussed, as it may impact long-term patient outcomes. However, none of these previous studies reported and compared the outcomes of all three techniques together.

The objective of this study was to compare the postoperative outcomes of all three CABG techniques to determine which technique(s) are most appropriate for the patients in the aspect of postoperative ischemic injury, hemodynamic function, and long-term survival.

## 2. Materials and methods

### 2.1 Patients and setting

This therapeutic research was conducted with a retrospective study observational cohort design conducted at Maharaj Nakorn Chiang Mai Hospital, Faculty of Medicine, Chiang Mai University, Thailand, with approval from the Research Ethics Committee Faculty of Medicine,

Chiang Mai University (SUR-2562-06606, No.334/2019). The target population was patients with coronary artery disease who underwent one of three different CABG procedures (ONCAB, OPCAB, and ONBHCAB) from January 2009-December 2020. The inclusion criteria were patients older than 18 years of age with coronary artery disease who had indications for CABG without other concomitant surgery. Patients whose operative techniques were converted intraoperatively from OPCAB to on-pump CABG were excluded from this study. And the patients who present with pre-operative acute coronary syndrome, acute myocardial infarction, pre-operative shock, and emergency CABG were excluded from this study.

Patients were divided into three groups based on the operative techniques used. Patients' selection of techniques was based on the surgeon's preference, and all six surgeons in the study perform CABG using all three techniques. However, the volume of each technique varied among the surgeons. To account for this, the surgeon variable was included as a confounding factor in the statistical analysis (detail are provided in the statistical analysis section).

## 2.2 Operative techniques

**2.2.1 OPCAB.** After conduit harvesting, heparin 1.5 mg/kg was administrated to patients and maintained the activated clotting time of> 300 seconds. The bypass grafting was performed with augmentation by the heart positioner and stabilizer. Each conduit graft was evaluated for the quality of bypass grafting by using transit time flow measurement.

**2.2.2 ONBHCAB and ONCAB.** In these two groups, CPB was performed using a Stockert Roller pump (Stockert Instrument Gmb H) and Inspire8 (LivaNova, Germany) oxygenator. Initially, a 3 mg/kg dose of heparin was administered intravenously with subsequent doses adjusted to maintain an activated clotting time > 400 seconds. During CPB, blood flow was maintained within a 2.4 to 3.0 L/min/m$^2$ range and in a normothermic state. A target hematocrit level between 20–25% was maintained under CPB. In ONBHCAB, the bypass grafting was performed in the same manner as in the OPCAB technique.

During ONCAB, after aortic cross-clamping, a myocardial protection strategy was performed via Buckberg's blood cardioplegia with an initial dose of the total volume of 1,800 ml and repeated with an administration of 550 ml every 20 minutes during bypass grafting. The aortic cross-clamp was removed after completion of the distal and proximal anastomoses.

## 2.3 Data collections

All patients underwent intensive cardiac output monitoring using a continuous cardiac output/oximetry/volumetric monitor: Vigilance II monitor (Edwards Lifesciences Corporation, USA) via Swan Ganz's catheter. The hemodynamic parameters were recorded after finished the operation for the intra-operative period. Venous blood samples were taken for cardiac enzyme test, and hemodynamics parameters were recorded again after patients arrived in ICU, and 24 hours post-operation.

## 2.4 Endpoints

The primary endpoints were displayed in three issues:

Firstly, post-operative myocardial injury focused on cardiac enzyme level; secondly, hemodynamic outcomes focused on the postoperative cardiac index (CI), mean arterial pressure (MAP), and use of inotropes; thirdly, long-term outcome focused on survival rate.

To measurement of myocardial injury, specific markers for ischemic myocardial injury were creatine kinase-MB (CK-MB) (mcg/L) and Troponin T (cTnT) (ng/l). The level of Troponin T and CK-MB usually reach their peak level around 24 hours after CABG. The level of both cardiac enzymes, immediately and 24 hours were used for comparing the myocardial

injury. The trends of the two enzymes were quite similar, but CK-MB reached their peak earlier than the cTnT [11]. The interval changes of markers use to infer ischemic injury during the corresponding interval time.

For the hemodynamic function, the following parameters were examined: the Cardiac Index (CI) to represent heart functionality; in addition to mean arterial pressure (MAP), mean pulmonary artery pressure (MPAP), and systemic vascular resistance index (SVRI). Postoperative use of the inotropic drugs was also monitored to facilitate a comparison of postoperative hemodynamic function among the patient groups.

Long-term outcomes were monitored for up to 12 years after surgery, focusing on survival from cardiac-related death and major adverse cardiovascular events (MACE), which included the composite of total death, myocardial infarction, coronary revascularization, stroke, and heart failure.

The secondary endpoints were focused on postoperative adverse events, including arrhythmia, intra-aortic balloon use, myocardium infraction, stroke, acute kidney injury, early reoperation, and hospital death.

## 2.5 Statistical analysis

Statistical analyses were performed using STATA software, version 16.1. The sample size was calculated by test comparing two independent means based on previous studies resembling the design of this study [12]. Categorical data were described as frequencies and percentages and a chi-square test was applied to group comparisons. Continuous data was represented by a mean and standard deviation (SD), and analysis of variance (ANOVA) with Bonferroni was applied for comparisons between the three groups.

The differences in preoperative characteristics between all groups were eliminated by matching with a propensity score. A propensity score of multiple arms (3 groups), or the predicted probability of receiving ONCAB, OPCAB, and ONBHCAB, was calculated from the multinomial logistic regression model [13]. The variables included in the model for propensity score were age, sex, New York Heart Association Functional Class, Canadian Cardiovascular Society Classification, the use of angiotensin-converting enzyme inhibitor drugs/angiotensin receptor blockers drugs, the use of pre-operative Aspirin, the pre-operative use of Clopidogrel, end-stage renal disease, coronary artery disease type, and left main coronary disease. Standardized differences (STD) were used to compare between groups after matching, while STD>0.1 indicated there were differences between the groups. The variables measuring STD>0.1, including weight, pre-operative ejection fraction, nitrate use, and calcium channel blocker use, were used for further calculation adjustments. Because the surgeon variable was a confounding factor and might have a large effect on the outcomes. The number of cases that the surgeon had performed represents the experience of CABG or the experience of each technique. Thus, surgeons were adding to the equation for statistic calculation as the adjustment variable.

A comparison of continuous data variables between the groups in overall time-interval change was calculated by a repeated measure mixed-effects regression and adjusted by the STD>0.1 variables. The differences were presented with time-interval change.

The hazard ratio of cardiac-related mortality, MACE, estimated long-term survival curves, and estimated freedom from MACE among patients who underwent three different types of CABG using flexible parametric survival regression (STPM2) with time-varying coefficient (TVC). The incomplete revascularization was used to adjust to long-term survival. Post-operative inotropic drug use and adverse events risk ratios were analyzed by exploratory univariable and multivariable binomial (risk) regression.

All statistical differences were considered significant at P<0.05.

## 3. Results

### 3.1 Demographic and operative data

A total of 2,028 cases were included in this study. Patients were divided into three groups; 916 (45.2%) patients underwent ONCAB, 517 (25.5%) patients underwent OPCAB, and 595 (29.3%) patients underwent ONBHCAB. After matching with propensity scores, the number of each study group totaled 443 patients. (Table 1) All data were similar between groups. However, several variables with an STD >0.1 were weight, pre-operative ejection fraction, and use of nitrate and calcium channel blockers (S1 Table & S1 Fig).

Operative data and post-operative data are shown in Table 2. Operative time from shortest to longest was ONBHCAB followed by OPCAB and ONCAB. The cardiopulmonary bypass times were shorter in ONBHCAB than in ONCAB. The number of patients who received more than 3 coronary anastomoses was lower in OPCAB followed by ONCAB and ONBH-CAB. No difference in incomplete revascularization between the three groups.

### 3.2 Cardiac enzyme

When cardiac enzyme levels were compared at immediate post-op and 24 hours post-op, ONCAB had the highest CK-MB and cTnT levels. OPCAB was the lowest of both CK-MB and cTnT levels. (Fig 1 & Table 2) From the results, OPCAB was the technique that had lower myocardium ischemic injury than ONBHCAB and ONCAB.

Regarding overall time-interval change, there was a significant difference in the change of both CK-MB ($p < 0.001$) and cTnT levels ($p = 0.013$) between OPCAB, ONBHCAB, and ONCAB. For CK-MB, it was interpreted as a decrease in CK-MB level throughout the follow-up period in OPCAB, ONBHCAB, and ONCAB by 18.5, 8.3, and 20.2 mcg/L, respectively. For cTnT, it was interpreted as an increase in cTnT levels throughout the follow-up period in OPCAB, ONBHCAB, and ONCAB by 13.2, 133, and 3.6 ng/ml, respectively (Fig 1 & S2 Table).

### 3.3 Hemodynamic & inotropic drug use

CI, the most appropriate hemodynamic illustrated value, significantly increased in all three groups. However, there was no significant difference ($p = 0.158$) when compared between groups (Fig 2 & S2 Table).

MAP showed no significant differences in the change between all groups ($p = 0.744$). Upon analysis, MPAP had significant differences in the change between all groups ($p = 0.026$). MPAP could be interpreted as a decrease throughout the follow-up period in OPCAB, ONBH-CAB, and ONCAB respectively. Overall, these two hemodynamic data measurements were not strong factors to represent the hemodynamic function when compared to CI and SVRI.

The SVRI value decreased from immediate post-operative to 24 hours post-op in all groups but there were no significant differences. Mean SVRI values were highest in the ONCAB group in every period of follow-up. When comparing the change in SVRI, OPCAB was the technique with the highest SVRI decrease (Fig 2).

Multivariable risk regression analysis for post-op inotropic use showed that OPCAB significantly reduced the use of adrenaline ($p = 0.037$), and milrinone ($p < 0.001$), when compared to ONBHCAB and ONCAB (Table 3).

### 3.4 Postoperative adverse events

ONBHCAB had a significantly increased risk of acute kidney injury ($p = 0.049$) when compared to OPCAB and ONCAB. There were no significant differences in postoperative

**Table 1. Patient characteristics.**

| Variables | All patients | | | | Post-matched with PS score | | | |
|---|---|---|---|---|---|---|---|---|
| | OPCAB (n = 517) | ONBHCAB (n = 595) | ONCAB (n = 916) | p | OPCAB (n = 443) | ONBHCAB (n = 443) | ONCAB (n = 443) | p |
| Male, n (%) | 332 (64.2) | 377 (63.3) | 546 (59.6) | 0.154 | 284 (64.11) | 281 (63.4) | 278 (62.8) | 0.920 |
| Age (Year, Mean ± SD) | 63.5 ±9.9 | 64.1 ±8.9 | 62.9 ±8.3 | 0.042 | 64.0 ±9.3 | 63.7 ±8.9 | 63.8 ±8.5 | 0.903 |
| Weight (kg, Mean ± SD) | 61.3 ±12.1 | 60.8 ±11.7 | 60.0 ±11.6 | 0.101 | 60.9 ±11.8 | 60.9 ±11.7 | 59.6 ±11.6 | 0.116 |
| Height (cm, Mean ± SD) | 158.7 ±8.5 | 158.5 ±9.0 | 157.8 ±9 | 0.116 | 158.7 ±8.5 | 158.4 ±9.1 | 158.2 ±8.4 | 0.772 |
| NYHA FC, n (%) | | | | 0.020 | | | | 0.994 |
| • 1 | 172 (33.2) | 148 (24.8) | 288 (31.4) | | 139 (31.4) | 132 (29.8) | 134 (30.3) | |
| • 2 | 273 (52.8) | 347 (58.3) | 508 (55.5) | | 239 (53.9) | 251 (56.6) | 247 (56.7) | |
| • 3 | 66 (12.7) | 85 (14.3) | 101 (11.0) | | 60 (13.5) | 55 (12.4) | 57 (12.9) | |
| • 4 | 6 (1.2) | 15 (2.5) | 19 (2.1) | | 5 (1.1) | 5 (1.1) | 5 (1.1) | |
| CCS, n (%) | | | | 0.011 | | | | 0.982 |
| • 0 | 46 (8.9) | 62 (10.4) | 138 (15.1) | | 42 (9.5) | 39 (8.8) | 48 (10.8) | |
| • 1 | 302 (58.4) | 332 (55.8) | 461 (50.3) | | 251 (56.6) | 251 (56.6) | 250 (56.4) | |
| • 2 | 128 (24.7) | 155 (26.1) | 236 (25.7) | | 114 (25.7) | 119 (26.8) | 114 (25.7) | |
| • 3 | 33 (6.4) | 35 (5.9) | 56 (6.1) | | 28 (6.3) | 28 (6.3) | 26 (5.9) | |
| • 4 | 8 (1.6) | 11 (1.9) | 25 (2.7) | | 8 (1.8) | 6 (1.3) | 5 (1.1) | |
| Pre-op LVEF (%, Mean ± SD) | 54.6 ±14.8 | 53.6 ±14.9 | 53.4 ±14.1 | 0.336 | 54.9 ±14.7 | 54.4 ±14.6 | 52.5 ±14.2 | 0.053 |
| Medication, n (%) | | | | | | | | |
| • Beta-Blocker | 359 (69.4) | 408 (68.6) | 671 (73.3) | 0.100 | 310 (69.8) | 298 (67.3) | 319 (72.1) | 0.311 |
| • ACEI & ARBs | 272 (52.6) | 286 (48.1) | 515 (56.2) | 0.008 | 234 (52.8) | 232 (52.4) | 241 (54.4) | 0.823 |
| • Nitrate | 356 (68.8) | 396 (66.6) | 589 (64.3) | 0.208 | 313 (70.7) | 285 (64.3) | 283 (63.9) | 0.058 |
| • Statin | 420 (81.2) | 471 (79.1) | 752 (82.1) | 0.358 | 362 (81.7) | 349 (78.8) | 354 (79.9) | 0.542 |
| • CCB | 102 (19.7) | 92 (15.4) | 168 (18.3) | 0.155 | 95 (21.4) | 71 (16.0) | 83 (18.7) | 0.122 |
| • Aspirin | 411 (79.5) | 483 (81.1) | 784 (85.6) | 0.006 | 360 (81.3) | 358 (80.8) | 363 (81.9) | 0.918 |
| • Clopidogrel | 314 (60.7) | 388 (65.2) | 621 (67.8) | 0.027 | 277 (62.5) | 263 (59.4) | 276 (62.3) | 0.575 |
| DM type2, n (%) | 244 (47.2) | 283 (47.6) | 445 (48.6) | 0.857 | 212 (47.8) | 213 (48.1) | 202 (45.6) | 0.718 |
| Hypertension, n (%) | 434 (83.9) | 504 (84.7) | 788 (89.0) | 0.533 | 376 (84.8) | 372 (83.9) | 375 (84.6) | 0.945 |
| Dyslipidemia, n (%) | 361 (69.8) | 422 (70.9) | 681 (74.3) | 0.131 | 312 (70.4) | 317 (71.5) | 321 (72.4) | 0.810 |
| Old CVA, n (%) | 17 (3.3) | 27 (4.5) | 40 (4.4) | 0.538 | 17 (3.8) | 17 (3.8) | 22 (5.0) | 0.661 |
| Previous PCI, n (%) | 32 (6.2) | 37 (6.2) | 46 (5.0) | 0.505 | 30 (6.7) | 29 (6.6) | 23 (5.2) | 0.598 |
| CKD, n (%) | 137 (26.5) | 182 (30.6) | 237 (25.8) | 0.118 | 117 (26.4) | 122 (27.5) | 116 (26.2) | 0.892 |
| ESRD, n (%) | 22 (4.3) | 40 (6.7) | 32 (3.5) | 0.015 | 17 (3.8) | 19 (4.3) | 19 (4.3) | 0.954 |
| Pre-op Creatinine (mg/dL, Mean ± SD) | 1.4 ±1.3 | 1.5 ±1.6 | 1.3 ±1.0 | 0.013 | 1.4 ±1.4 | 1.4 ±1.3 | 1.4 ±1.2 | 0.699 |
| CAD type, n (%) | | | | <0.001 | | | | 0.994 |
| • 1vv | 23 (4.5) | 4 (0.7) | 9 (1.0) | | 4 (0.9) | 4 (0.9) | 4 (0.9) | |
| • 2vv | 101 (19.5) | 66 (11.1) | 126 (13.8) | | 65 (14.7) | 62 (14.0) | 60 (13.6) | |
| • 3vv | 393 (76.0) | 525 (88.2) | 781 (85.2) | | 374 (84.4) | 377 (85.1) | 379 (85.6) | |
| LM disease, n (%) | 276 (53.4) | 286 (48.1) | 405 (44.2) | 0.004 | 237 (53.5) | 235 (53.1) | 238 (53.7) | 0.985 |

OPCAB, Off-pump coronary artery bypass; ONBHCAB, On-pump beating heart coronary artery bypass; ONCAB, On-pump arrested heart coronary artery bypass; NYHA FC, New York Heart Association functional classification; CCS, Canadian Cardiovascular Society Classification; Pre-op, Pre-operative; LVEF, Left ventricular ejection fraction; ACEI/ARBs, Angiotensin-converting enzyme inhibitor drugs/Angiotensin receptor blockers drugs; CCB, Calcium channel blocker; DM, Diabetes Miletus; CVA, Cerebrovascular disease; PCI, percutaneous cardiac intervention; CKD, Chronic kidney disease; ESRD, End-stage renal disease; CAD, Coronary artery disease; vv, vessel disease; LM, Left main.

Statistically significant at $p<0.05$.

**Table 2. Operative and post-operative data.**

| Variable | OPCAB (n = 443) | ONBHCAB (n = 443) | ONCAB (n = 443) | p-value |
|---|---|---|---|---|
| Operative data | | | | |
| Operative time (min, Mean ± SD) | 250.1 ±91.1 | 247 ±61.8 | 305 ±82.1 | <0.001 |
| CPB time (min, Mean ± SD) | - | 72.7 ±28.9 | 129.2 ±86.0 | <0.001 |
| Coronary anastomosis (Mean ± SD) | 4.0 ±1.2 | 4.3 ±1.1 | 4.3 ±1.1 | 0.295 |
| Grafted Coronary Artery Target > 3, n (%) | 300 (68.9) | 359 (81.0) | 342 (77.2) | <0.001 |
| Incomplete revascularization, n (%) | 19 (4.5) | 36 (8.4) | 34 (7.9) | 0.065 |
| Cardiac enzyme | | | | |
| CK-MB, (Mean ± SD) | | | | |
| • Immediate post-op | 25.4 ±15.14 | 53.9 ±27.5 | 64.7 ±78.9 | <0.001 |
| • 24 hr. post-op | 25.9 ±24.7 | 34.4 ±40.6 | 35.3 ±24.1 | <0.001 |
| cTnT, (Mean ± SD) | | | | |
| • Immediate post-op | 369.7 ±582.1 | 571.5 ±575.2 | 834.6 ±734.4 | <0.001 |
| • 24 hr. post-op | 517.9 ±747.7 | 655.8 ±775.2 | 799.0 ±927.7 | 0.001 |
| Hemodynamic | | | | |
| CI, (Mean ± SD) | | | | |
| • Immediate post-op | 2.26 ±0.51 | 2.31 ±0.51 | 2.28 ±0.53 | 0.443 |
| • 24 hr. post-op | 2.51 ±0.51 | 2.46 ±0.44 | 2.47 ±0.49 | 0.307 |
| MAP, (Mean ± SD) | | | | |
| • Immediate post-op | 81.0 ±12.4 | 82.9 ±12.7 | 83.0 ±13.5 | 0.031 |
| • 24 hr. post-op | 78.5 ±10.0 | 79.1 ±9.8 | 79.3 ±10.0 | 0.495 |
| MPAP, (Mean ± SD) | | | | |
| • Immediate post-op | 23.0 ±5.88 | 23.0 ±6.24 | 22.1 ±6.14 | 0.056 |
| • 24 hr. post-op | 20.3 ±6.16 | 21.0 ±6.14 | 20.5 ±6.54 | 0.316 |
| SVRI, (Mean ± SD) | | | | |
| • Immediate post-op | 2499 ±747 | 2521 ±762 | 2540 ±818 | 0.760 |
| • 24 hr. post-op | 2245 ±547 | 2334 ±569 | 2351 ±581 | 0.021 |
| Post-operative data | | | | |
| 24 hours mediastinal drainage (ml, Mean ± SD) | 425.1 ±245 | 460.4 ±284 | 424.3 ±289 | 0.091 |
| Post-op Inotropic drug used, n (%) | | | | |
| • Norepinephrine | 180 (40.6) | 170 (38.4) | 157 (35.4) | 0.277 |
| • Adrenaline | 7 (1.6) | 15 (3.4) | 34 (7.7) | <0.001 |
| • Dobutamine | 107 (24.1) | 96 (21.8) | 110 (24.8) | 0.516 |
| • Milrinone | 17 (3.8) | 24 (5.4) | 38 (8.6) | 0.012 |
| Post-op Arrythmia, n (%) | 106 (23.9) | 102 (23.2) | 110 (24.8) | 0.829 |
| Post-op IABP, n (%) | 52 (12.2) | 64 (14.8) | 53 (12.2) | 0.445 |
| Post-op MI, n (%) | 2 (0.5) | 2 (0.5) | 1 (0.3) | 1.000 |
| Stroke, n (%) | 3 (0.7) | 3 (0.7) | 5 (1.1) | 0.803 |
| Acute kidney injury, n (%) | 17 (3.8) | 27 (6.1) | 27 (6.1) | 0.228 |
| Post-op New Hemodialysis, n (%) | 1 (0.2) | 2 (0.5) | 4 (0.9) | 0.518 |
| Early Re-operation, n (%) | 6 (1.4) | 7 (1.6) | 11 (2.5) | 0.544 |
| ICU stay (Day, Mean ± SD) | 1.3 ±1.1 | 1.4 ±1.7 | 1.4 ±1.2 | 0.366 |
| Hospital stay (Day, Mean ± SD) | 12.2 ±6.9 | 11.8 ±9.9 | 12.7 ±7.7 | 0.294 |
| Hospital death (Early), n (%) | 3 (0.7) | 6 (1.4) | 9 (2.1) | 0.218 |
| Coronary re-revascularization, n (%) | 5 (1.1) | 7 (1.6) | 8 (1.8) | 0.701 |

(*Continued*)

**Table 2.** (Continued)

| Variable | OPCAB (n = 443) | ONBHCAB (n = 443) | ONCAB (n = 443) | p-value |
|---|---|---|---|---|
| Re-admit heart failure, n (%) | 57 (13.6) | 68 (15.8) | 73 (16.9) | 0.377 |

OPCAB, Off-pump coronary artery bypass; ONBHCAB, On-pump beating heart coronary artery bypass; ONCAB, On-pump arrested heart coronary artery bypass; CPB, Cardiopulmonary bypass; CK-MB, creatine kinase-MB (mcg/L); cTnT, Cardiac Troponin T (ng/ml); CI, Cardiac Index (L/min.m2); MAP, Mean arterial pressure (mm Hg); MPAP, Mean pulmonary artery pressure (mm Hg); SVRI, Systemic vascular resistant index (dn.s.m2/cm5); Immediate, Immediate post-operative; 24 hr., 24 hours post-operative; Intra-op, Intra-operative; Post-op, Post-operative; IABP, Intra-aortic balloon pump; Post-op MI, Post-operative myocardial infarction; ICU, Intensive care unit.

Statistically significant at $p<0.05$.

arrhythmia, IABP, MI, stroke, early re-operation, or hospital death between the three groups (Table 4).

### 3.5 Survival from cardiac-related death and free from MACE

Five-year survival for OPCAB, ONBHCAB, and ONCAB were 94.8%, 96.4%, and 89.9%, respectively. And ten-year survival for OPCAB, ONBHCAB, and ONCAB were 80.5%, 75.9%, and 73.7%, respectively. Five-year freedom from MACE for OPCAB, ONBHCAB, and ONCAB were 86.53%, 86.63%, and 76.83%, respectively. And ten-year freedom from MACE was 66.30%, 52.5%, and 45.55%, respectively.

Flexible parametric survival analysis found that OPCAB was associated with a significant reduction of mortality risk (hazard ratio (HR) OPCAB = 0.33, p = 0.001, 95%CI 0.17–0.65,) (Fig 3) and significant reduction of MACE (HR OPCAB = 0.52, p = 0.004, 95%CI 0.34–0.81) (Fig 4). A post-matched crude analysis by Kaplan Meier survival graph is shown in S2 Fig.

## 4. Discussion

When comparing three techniques, short-term results showed that OPCAB had the least impact on myocardium injury and no significant difference in postoperative hemodynamic

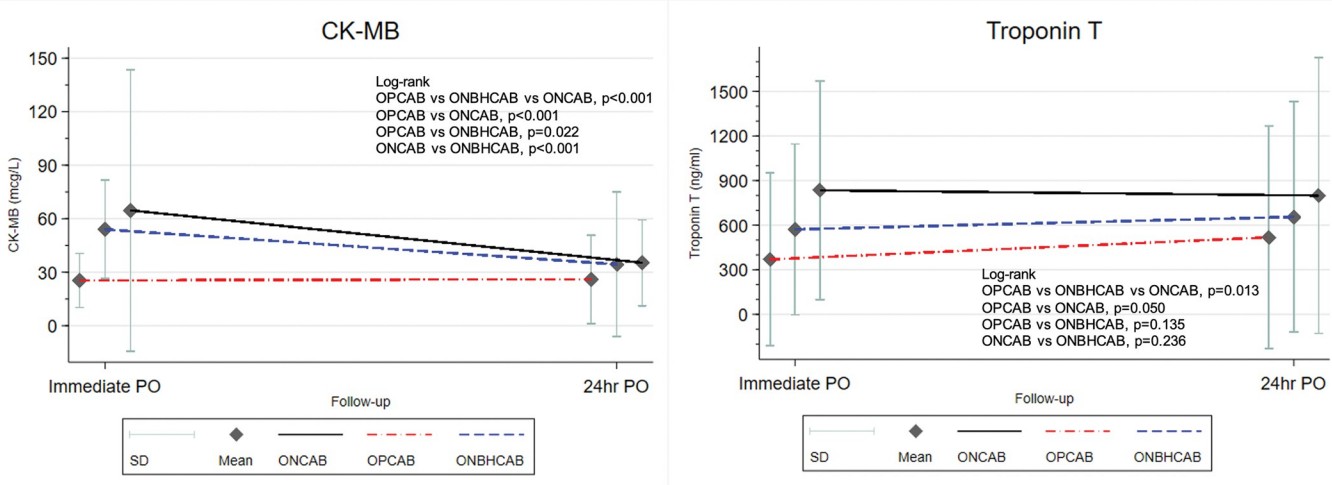

**Fig 1. Graft for the time interval changes of cardiac enzyme between three groups.** OPCAB, Off-pump coronary artery bypass; ONBHCAB, On-pump beating heart coronary artery bypass; ONCAB, On-pump arrested heart coronary artery bypass; CK-MB, creatine kinase-MB (mcg/L); cTNT, Cardiac Troponin T (ng/ml). Statistically significant at $p<0.05$.

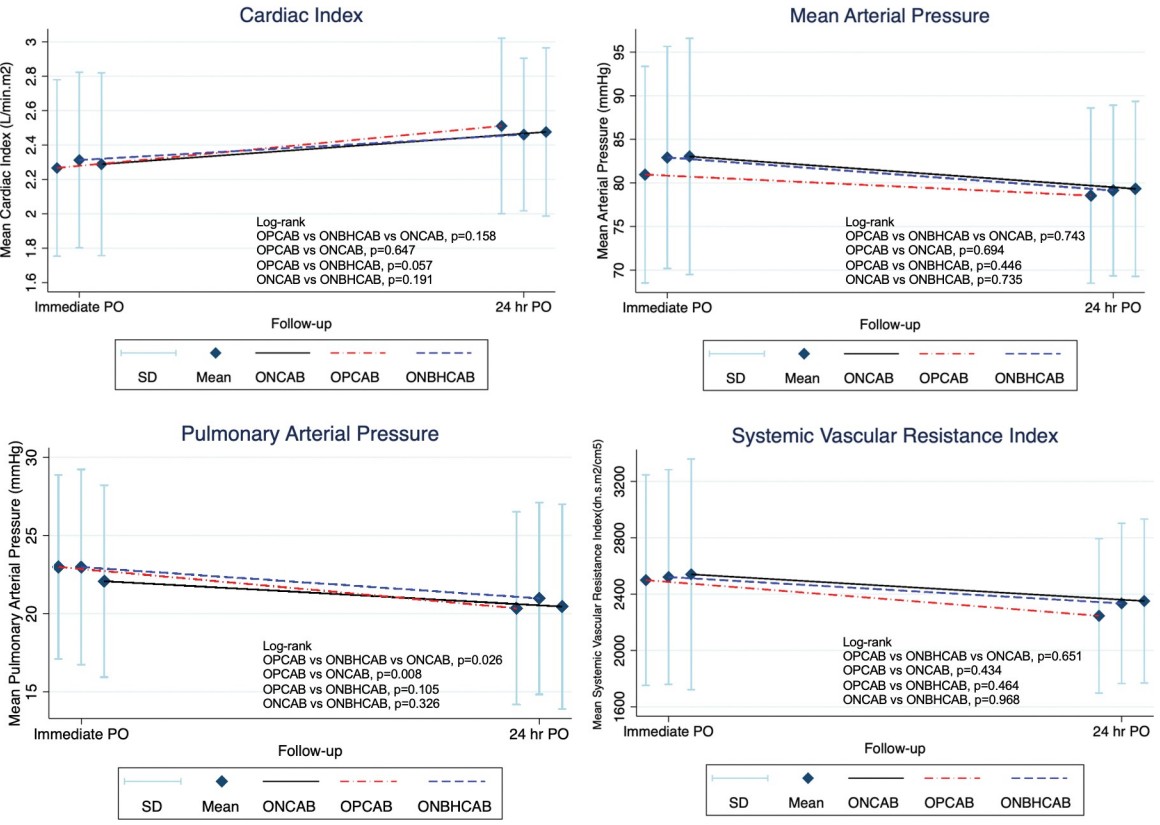

**Fig 2. Graft for the time interval changes of hemodynamic values between the three groups.** OPCAB, Off-pump coronary artery bypass; ONBHCAB, On-pump beating heart coronary artery bypass; ONCAB, On-pump arrested heart coronary artery bypass; CI, Cardiac Index (L/min.m2); MAP, Mean arterial pressure (mm Hg); MPAP, Mean pulmonary artery pressure (mm Hg); SVRI, Systemic vascular resistant index (dn.s.m2/cm5); Changed, Time interval changed. Statistically significant at $p<0.05$.

function compared to other techniques. Additionally, ONBHCAB was associated with an increased risk of kidney injury in the postoperative period. Long-term results found that OPCAB had the highest for long-term overall survival, freedom from MACE, significantly reduced mortality risk, and reduce MACE compared to other techniques.

The rising CK-MB and cTnT levels represented ischemic myocardium injury. In this study, the ONCAB group had the highest CK-MB and cTnT in the immediate and 24-hour postoperative period, while the OPCAB group had the lowest levels when compared with all three groups. Previous studies by Alwan et al. [12] and Asciwan et al. [14] reported significantly higher troponin levels in patients who underwent conventional CABG compared to OPCAB. Rastan et al. compared ONBHCAB and OPCAB and found that ONBHCAB had significantly higher CK-MB and troponin I levels. They concluded CPB may have an adverse effect on the myocardium and that OPCAB caused less myocardial injury [10].

Due to the need to arrest the heart during ONCAB, a cardioplegic solution was administered, resulting in patient hypothermia. Global ischemic events could have occurred during ONCAB while patients undergoing OPCAB had only local ischemia when the coronary artery was occluded to perform the anastomosis. Even ONBHCAB patients had more myocardial injuries than OPCAB patients, likely due to CPB still having an adverse effect on the myocardium despite the heart not being arrested. D Hert and colleagues explained the pathophysiology related to CPB, which includes provoked a systemic inflammatory response, depletion of high-energy phosphates, and disturbance of the intracellular calcium, all of which can damage

**Table 3. Postoperative inotropic drug used.**

|  | RR | P | 95%CI |
|---|---|---|---|
| All Inotropic drug used [1] |  |  |  |
| • OPCAB | 0.99 | 0.933 | 0.88, 1.12 |
| • ONBHCAB | 0.92 | 0.223 | 0.81, 1.04 |
| • ONCAB | reference |  |  |
| Adrenaline [2] |  |  |  |
| • OPCAB | 0.36 | 0.037 | 0.13, 0.93 |
| • ONBHCAB | 0.60 | 0.278 | 0.24, 1.50 |
| • ONCAB | reference |  |  |
| Norepinephrine [3] |  |  |  |
| • OPCAB | 0.99 | 0.950 | 0.79, 1.23 |
| • ONBHCAB | 1.17 | 0.219 | 0.90, 1.51 |
| • ONCAB | reference |  |  |
| Dobutamine [4] |  |  |  |
| • OPCAB | 0.85 | 0.224 | 0.67, 1.10 |
| • ONBHCAB | 0.79 | 0.074 | 0.61, 1.02 |
| • ONCAB | reference |  |  |
| Milrinone [5] |  |  |  |
| • OPCAB | 0.30 | <0.001 | 0.16, 0.56 |
| • ONBHCAB | 0.89 | 0.727 | 0.47, 1.67 |
| • ONCAB | reference |  |  |

Significant exploratory univariable of inotropic variables were used for multivariable as.

1 Adjusted by age, weight, LVEF, ESRD, left main disease.

2 Adjusted by LVEF, ESRD, operative time, surgeon.

3 Adjusted by sex, clopidogrel, left main disease, surgeon.

4 Adjusted by age, sex, weight, height, LVEF, calcium channel blocker, left main disease, operative time.

5 Adjusted by weight, LVEF, NYHA FC, ESRD, coronary artery disease type, operative time, surgeon.

OPCAB, Off-pump coronary artery bypass; ONBHCAB, On-pump beating heart coronary artery bypass; ONCAB, On-pump arrested heart coronary artery bypass; RR, Risk Ratio; LVEF, Left ventricular ejection fraction; NYHA FC, New York Heart Association functional classification; ESRD, End stage renal disease.

Statistically significant at p <0.05.

the myocardium [15]. On 24-hours follow-up for the cardiac enzyme, the change time-interval for the cardiac enzyme was not as strong an indicator of myocardial injury as their original value.

In recent times, new biomarkers have emerged that can be compared to cardiac troponin, the current 'gold standard' surrogate biomarker of myocardial damage. Plasma exosomes are small extracellular vesicles that are released into the bloodstream and contain various molecular cargoes that play a role in intercellular communication. During cardiac surgery, ischemia-reperfusion injury can lead to tissue damage and inflammation, resulting in exosome release from injured or stressed cells [16]. Emanueli et al. studied exosomes containing cardiac micro-RNAs and found that plasma concentrations of exosomes and their cargo of cardiac micro-RNAs increased in patients undergoing CABG and correlated with cardiac troponin levels [17]. Carrozzo et al. studied between plasma exosomes and serum cardiac troponin I levels in older (on-CPB) CABG patients. After aortic de-clamping, exosome levels significantly increased at both 3 hours and 72 hours, while troponin I level peaked at 3 hours and then gradually decreased [18]. Moreover, Frey et al. discovered that exosome release was related to ischemic/reperfusion injury and associated with myocardial tissue protection [19]. Therefore,

**Table 4. Post-operative adverse event.**

| | RR | *P* | 95%CI |
|---|---|---|---|
| Post-operative Arrythmia [1] | | | |
| • OPCAB | 1.21 | *0.143* | 0.92, 1.67 |
| • ONBHCAB | 0.87 | *0.464* | 0.61, 1.25 |
| • ONCAB | reference | | |
| IABP post-op [2] | | | |
| • OPCAB | 1.19 | *0.323* | 0.84, 1.72 |
| • ONBHCAB | 1.30 | *0.140* | 0.92, 1.85 |
| • ONCAB | reference | | |
| Post-op MI [3] | | | |
| • OPCAB | 1.31 | 0.857 | 0.06–25.8 |
| • ONBHCAB | 1.53 | 0.805 | 0.05–44.40 |
| • ONCAB | reference | | |
| Stroke [4] | | | |
| OPCAB | 0.75 | *0.694* | 0.19, 3.06 |
| ONBHCAB | 0.89 | *0.878* | 0.21, 3.75 |
| ONCAB | reference | | |
| Acute kidney injury [5] | | | |
| • OPCAB | 1.23 | *0.556* | 0.61, 2.50 |
| • ONBHCAB | 1.95 | *0.049* | 1.02, 4.42 |
| • ONCAB | reference | | |
| Early Re-operation [6] | | | |
| • OPCAB | 0.54 | *0.230* | 0.20, 1.46 |
| • ONBHCAB | 0.62 | *0.324* | 0.24, 1.59 |
| • ONCAB | reference | | |
| Hospital death [7] | | | |
| • OPCAB | 0.32 | 0.092 | 0.08, 1.20 |
| • ONBHCAB | 0.67 | 0.439 | 0.24, 1.85 |
| • ONCAB | reference | | |

Significant exploratory univariable of post-operative adverse event variables were used for multivariable as.

1 Adjusted by age, weight, LVEF, NYHA FC, end-stage renal disease, operative time, surgeon.

2 Adjusted by age, weight, LVEF, NYHA FC, beta blocker, calcium channel blocker, statin, hypertension, dyslipidemia, left main disease, operative time, surgeon.

3 No variable adjustment.

4 Adjusted by end-stage renal disease, operative time.

5 Adjusted by LVEF, CCS, aspirin, clopidogrel, hypertension, dyslipidemia, operative time, surgeon.

6 Adjusted by previous PCI.

7 Adjusted by age.

OPCAB, Off-pump coronary artery bypass; ONBHCAB, On-pump beating heart coronary artery bypass; ONCAB, On-pump arrested heart coronary artery bypass; RR, Risk Ratio; IABP, Intra-aortic balloon pump; Post-op MI, Post-operative myocardial infarction; LVEF, Left ventricular ejection fraction; NYHA FC, New York Heart Association functional classification; CCS, Canadian Cardiovascular Society Classification; Pre-op, Pre-operative; PCI, Percutaneous cardiac intervention.

Statistically significant at p <0.05.

due to their persistence and specific biomarkers for cardiac injury, exosomes have the potential to be a valuable new standard tool in the detection and monitoring of myocardial injury in the near future.

## Long-term survival

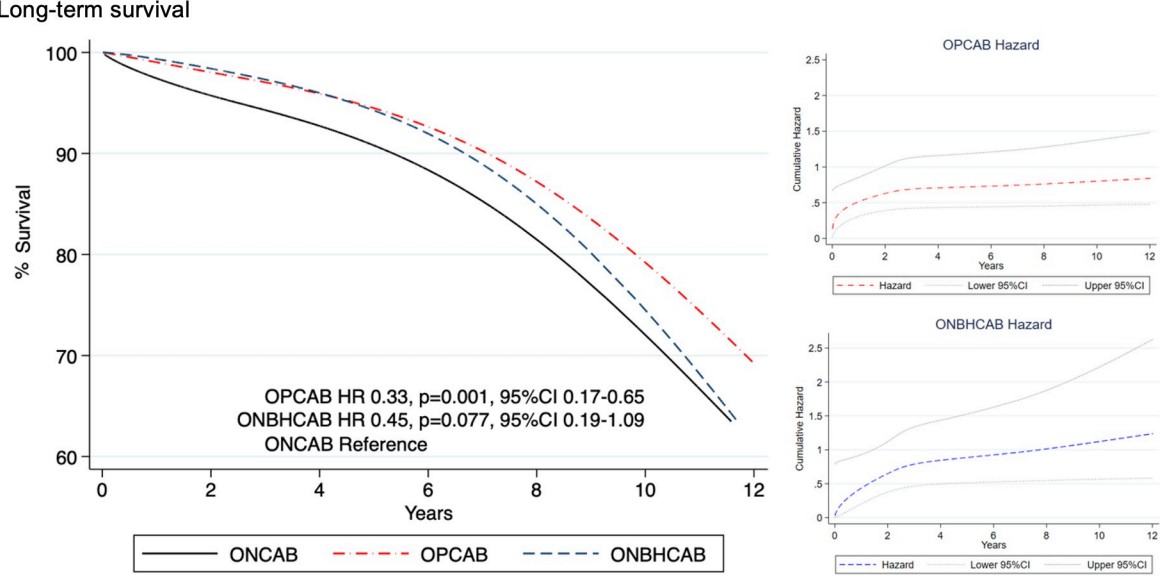

**Fig 3. Estimated long-term survival curves among patients who underwent three different types of CABG using flexible parametric survival regression with time-varying coefficient.** OPCAB, Off-pump coronary artery bypass; ONBHCAB, On-pump beating heart coronary artery bypass; ONCAB, On-pump arrested heart coronary artery bypass; HR, Hazard ratio.

According to the hemodynamic result, all CI values from each technique significantly increased in the postoperative period. However, when comparing the change of CI between all three groups, there were no differences between groups. This indicated that CI increased in all three techniques without significant differences after CABG. Postoperative cardiac functions improved because the ischemic myocardium received the oxygenated blood from bypass

## Freedom from major adverse cardiovascular events

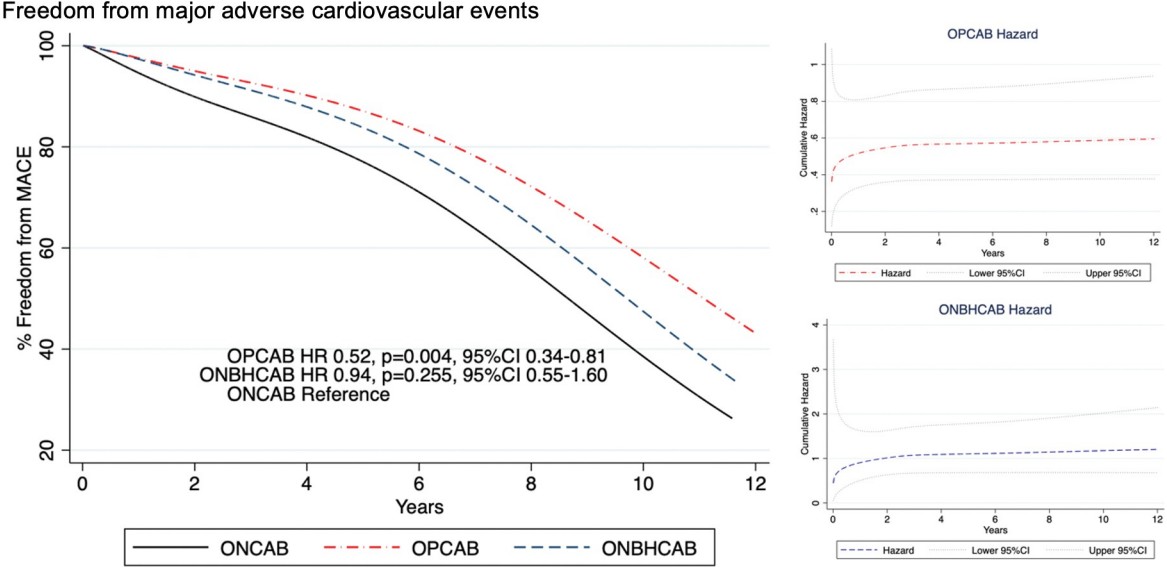

**Fig 4. Estimated MACE-free curves among patients who underwent three different types of CABG using flexible parametric survival regression with time-varying coefficient.** OPCAB, Off-pump coronary artery bypass; ONBHCAB, On-pump beating heart coronary artery bypass; ONCAB, On-pump arrested heart coronary artery bypass; HR, Hazard ratio; MACE, Major adverse cardiovascular events: The composite of total death, myocardial infarction, coronary revascularization, stroke, and heart failure.

grafting. Tatoulis et al. found no statistical differences in mean CI values, or in the pattern response during the first 24 hours between ONCAB and OPCAB [20]. Pegg and colleagues studied hemodynamic function between ONCAB and ONBHCAB and found no differences in CI in either group in the early postoperative period [21].

The changes in SVRI were the clinical representation of systemic inflammatory response, vasodilation, and low cardiac output syndrome (LCOS), after CABG. In this study, OPCAB had the significantly highest decrease in SVRI value when compared with ONBHCAB and ONCAB. Some studies attributed the inflammatory response to surgical trauma during OPCAB (graft harvesting, cardiac manipulation, microvascular clamp, compression form stabilizer) as having a higher effect than the cause from CPB [22]. Gaudino et al. reviewed that OPCAB was associated with a significant increase in inflammatory mediators compared with ONCAB [2]. Although a decrease in SVRI represented systemic inflammation, there was still an advantage from the reduction in afterload and preload that had a benefit for immediate post-op myocardium function [20,23].

However, it cannot be concluded that OPCAB had the highest inflammatory response, as the postoperative SVRI value is calculated and influenced by multiple factors, including hypothermia, ischemic-reperfusion injury, hemodynamic instability (including LCOS), and inotropic drug use. Moreover, the lower SVRI in the OPCAB group might be related to the intraoperative changes that differ from the CPB group. During OPCAB, patients were always kept warm with higher intravascular volume to prevent hypotension during cardiac manipulation, which might result in lower SVRI because of less hypothermia and vasoconstriction compared to those under CPB. The use of some inotropic drugs also caused vasoconstriction and increased SVRI. Despite the study's results of, ONCAB and ONBHCAB had a higher number of inotropic drug use, which may have caused higher mean SVRI.

OPCAB significantly reduced the use of postoperative adrenaline and milrinone, likely due to the avoidance of CPB and its associated adverse effects, such as SIRs and vasodilatation, which require inotropic support. The higher numbers of inotropic drug use in ONCAB and ONBHCAB might be attributed to the use of CPB. However, it is important to note that the use of only adrenaline and milrinone as an indicator of postoperative inotropic support may introduce bias, as other drugs may also be used. Hussain and colleagues found there were significantly fewer patients requiring lower doses of inotropic support in OPCAB than in ONCAB [24], and Rastan et al. found a lower number of patients requiring postoperative inotropic drugs in OPCAB than in ONBHCAB [10]. Nonetheless, it is not possible to conclude that OPCAB had a lower rate of postoperative inotropic use, as other factors may also influence the need for such support.

Postoperative adverse events were not statistically significant, except for ONBHCAB, which demonstrated a higher risk for acute kidney injury (AKI). Although many reports regarding postoperative renal function in CABG exist, conclusive answers remain elusive. Some meta-analyses revealed that OPCAB had lower incidences of postoperative AKI than ONCAB [25]. Ming Jen Chan and colleagues compared the incidence of AKI in all three CABG techniques and found higher incidences in ONBHCAB than in ONCAB and higher incidences in ONBHCAB than OPCAB [26]. All studies explained that the mechanism of post-CABG AKI was caused by SIRs and coagulopathy from the CPB. In addition, micro-embolisms occurred during aortic cannulation, and related consequences may have facilitated renal ischemia [25].

In this study, a higher incidence of AKI in ONBHCAB could be due to a higher distended heart, higher right atrial pressure, less venous return from the kidney, and less glomerular filtration function. OPCAB provides pulsatile and normal kidney perfusion pressure, which transfers to some degree of renal protection. For the ONCAB group, left ventricular (LV) venting might help to decompress the heart, cause less venous returning pressure for the kidney,

and promote good urine output. However, ONBHCAB groups didn't use LV venting, which might result in LV distention, especially when under high-flow CPB running. Also, mobilization of the heart under a single two-staged venous cannula can cause a decrease in venous drainage in some positions and cause higher right atrial retuning pressure for the inferior vena cava and renal veins.

Results of high-quality CABG studies remain controversial and differ regarding mortality and composite of long-term adverse outcomes. The studies between OPCAB and ONCAB showed that OPCAB had neither a different nor a lower rate of survival. The CORONARY trial, with a mean follow-up of 4.8 years, showed no difference in composited outcomes (including death, MI, repeat revascularization, and stroke) [6]. The GOPCABE trial also showed no difference in 12 months of composited outcomes [3]. While the ROOBY-FS trial showed a 15.2% rate of death, and 31% MACE at 5 years in the OPCAB group versus an 11.9% rate of death, and 27.1% MACE in the ONCAB [1]. They concluded that off-pump CABG led to lower rates of 5-year survival and event-free survival. In contrast, the studies between ONBHCAB and ONCAB or ONBHCAB and OPCAB showed ONBHCAB had a mostly lower rate of mortality [7,9].

This study demonstrated that OPCAB resulted in significantly decreased risks of cardiac-related death and MACE in long-term outcomes. The better outcomes in the OPCAB group could be attributed to several factors. First, OPCAB resulted in less adverse effects from ischemic-reperfusion injury and no systemic inflammatory response, which could lead to better survival outcomes, particularly in the first 2 years after surgery. Second, the quality of bypass grafting in OPCAB may have been superior. Although the number of grafts was not different among the three groups in this study, there was a trend toward higher rates of complete revascularization in the OPCAB group (Table 2), which could also have contributed to long-term survival. Third, OPCAB was performed under normal physiologic LV filling and on a beating heart, while LV was decompressed during an on-pump technique. This could promote graft flow, even if the quality of anastomosis was not perfect. However, this study did not include long-term imaging to assess graft patency, so it was difficult to conclude that the better survival rate in OPCAB group resulted solely from better graft quality.

Finally, there were several limitations of this study. The main limitation was that this study was conducted retrospectively, which may have introduced heterogeneities between the study groups that could influence the final outcomes. Although propensity score matching methods were used to correct confounding factors and adjust unequal baselines to enable proper comparison, some hidden factors may not have been included in the adjustment. Secondly, the number of patients in each group was reduced as a result of the matching method, with the ONCAB group experiencing a reduction of 49.1% in patient numbers. A further limitation is that the comparison between cardiac enzyme and hemodynamic values was the surrogate outcome for myocardial injury and hemodynamic function, not direct clinical outcomes. Also, the follow-up period for assessing cardiac enzymes, and hemodynamic function may have been too short. The duration of cardiac enzyme assessment was typically around 48–72 hours, and hemodynamic data may have fluctuated due to clinical status. If more frequent data was collected, the trend and the change in the values might have been different.

## 5. Conclusion

OPCAB resulted in lower levels of postoperative myocardial injury compared to other techniques, while no significant differences in postoperative hemodynamic function were observed among all three techniques. ONBHCAB was significantly associated with a higher rate of postoperative renal impairment. Therefore, OPCAB was found to be the preferable technique associated with long-term survival benefits due to a significant reduction in mortality and MACE.

## Supporting information

**S1 Fig. Dot charts compare Standardized differences between pre-matched and post-matched patients' baseline characteristics.** NYHA FC, New York Heart Association functional classification; CCS, Canadian Cardiovascular Society Classification; Pre-op, Pre-operative; LVEF, Left ventricular ejection fraction; ACEI/ARBs, Angiotensin-converting enzyme inhibitor drugs/Angiotensin receptor blockers drugs; CCB, Calcium channel blocker; DM, Diabetes Miletus; CVA, Cerebrovascular disease; PCI, percutaneous cardiac intervention; CKD, Chronic kidney disease; ESRD, End-stage renal disease; CAD, Coronary artery disease; LM, Left main.
(TIF)

**S2 Fig. Crude analysis of Kaplan-Meier survival and freedom from MACE between three groups.** OPCAB, Off-pump coronary artery bypass; ONBHCAB, On-pump beating heart coronary artery bypass; ONCAB, On-pump arrested heart coronary artery bypass; MACE, Major adverse cardiovascular events: The composite of total death, myocardial infarction, coronary revascularization, stroke, and heart failure.
(TIF)

**S1 Table. Standardized differences data between pre-matched and post-matched of patients' baseline characteristics.**
(DOCX)

**S2 Table. Surgeon detail.**
(DOCX)

**S3 Table. Postoperative cardiac enzyme and hemodynamic function analyzed by repeated measure mixed effects regression.**
(DOCX)

**S1 Data. The minimal data set used for analyzed.**
(XLSX)

## Acknowledgments

We would like to thank the staff of Chiang Mai University's English language team (CELT) for providing language editing assistance.

## Author Contributions

**Conceptualization:** Amarit Phothikun, Thitipong Tepsuwan.

**Data curation:** Amarit Phothikun, Apichat Tantraworasin, Phichayut Phinyo, Thitipong Tepsuwan.

**Formal analysis:** Amarit Phothikun, Apichat Tantraworasin, Phichayut Phinyo.

**Investigation:** Amarit Phothikun, Apichat Tantraworasin, Thitipong Tepsuwan.

**Methodology:** Amarit Phothikun, Apichat Tantraworasin, Phichayut Phinyo, Thitipong Tepsuwan.

**Project administration:** Amarit Phothikun.

**Resources:** Amarit Phothikun, Weerachai Nawarawong.

**Supervision:** Amarit Phothikun, Weerachai Nawarawong, Apichat Tantraworasin, Phichayut Phinyo, Thitipong Tepsuwan.

**Validation:** Amarit Phothikun, Apichat Tantraworasin, Phichayut Phinyo, Thitipong Tepsuwan.

**Visualization:** Amarit Phothikun, Weerachai Nawarawong, Apichat Tantraworasin, Phichayut Phinyo, Thitipong Tepsuwan.

**Writing – original draft:** Amarit Phothikun, Thitipong Tepsuwan.

**Writing – review & editing:** Amarit Phothikun, Weerachai Nawarawong, Apichat Tantraworasin, Phichayut Phinyo, Thitipong Tepsuwan.

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
