## [Decision Letter · Decision Letter 0]

14 Mar 2023

PONE-D-23-02554The Outcomes of Three Different Techniques of Coronary Artery Bypass Grafting: On-pump Arrested Heart, On-pump Beating Heart, and Off-pump.PLOS ONE

Dear Dr. Tepsuwan,

Thank you for submitting your manuscript to PLOS ONE. After careful consideration, we feel that it has merit but does not fully meet PLOS ONE’s publication criteria as it currently stands. Therefore, we invite you to submit a revised version of the manuscript that addresses the points raised during the review process.

ACADEMIC EDITOR: Although the manuscript addresses an interesting topic, it is affected by relevant shortcomings that require an in-depth revision. All issues are required.==============================

We look forward to receiving your revised manuscript.

Kind regards,

Vincenzo Lionetti, M.D., PhD

Academic Editor

PLOS ONE

Journal Requirements:

Reviewers' comments:

Reviewer's Responses to Questions

**Comments to the Author**

1. Is the manuscript technically sound, and do the data support the conclusions?

Reviewer #1: No

Reviewer #2: Yes

2. Has the statistical analysis been performed appropriately and rigorously? 

Reviewer #1: Yes

Reviewer #2: Yes

3. Have the authors made all data underlying the findings in their manuscript fully available?

Reviewer #1: No

Reviewer #2: No

4. Is the manuscript presented in an intelligible fashion and written in standard English?

Reviewer #1: Yes

Reviewer #2: Yes

5. Review Comments to the Author

Reviewer #1: Congratulations for the arguments that authors have selected.

Materials and Method

Reasoning behind the exclusion from off-pump to on-pump surgery?

What kind of propensity matching algorithm did you use? After matching form smallest group (517 OPCAB) 443 remained.

Results

the quantity of grafts in groups was different?

present P-value for survival differences.

Coronary anastomosis is also taking into account sequential grafting or only graft numbers?

Very interesting results. Re-fine a little bit your propensity matching technique, usually it is well defined for two groups. In the tables the p-values are post hoc corrected? If not, do so to have a realistic number of p-values.

Reviewer #2: i read with great interest the MS by Phothikun et al.

the authors concluded OPCAB implementation resulted in a lower occurrence of postoperative ischemic injury than ONCAB and ONBHCAB. No differences in postoperative hemodynamic function in all three techniques were observed. OPCAB respectively were preferable techniques beneficial for long-term outcomes.

the MS is technically sound but i have several comments

1. the authors conducted PS-matching however the process of variables selection is not well defined; i agree n of cases per surgeon should remain adjustment variable but other essential variables (like in Table 1) other than ACEi inhibitors intake should be considered;

2. Authors adjust long-term survival with completeness of revascularization which is correct; why not to PS-match the patients with the n. of anastomoses and/or CR

3. authors mention converted patients were excluded from the study, but how many there were? it seems from the total numbers, OPCAB is not preferred technique (25% of total); are there any factors other than surgeons' preference to drive patients towards one or another approach?

4. OPCAB is further performed predominantly by one surgeon, with higher rates of complete revascularization in this group, contra-intuitively, it poses a limitation; the results of improved survival and reduced ischemia can, therefore, be translatable to broader population of patients only provided that surgeons performing OPCAB have experience in OPCAB and can offer complete revascularization.

5. language editing is required

6. PLOS authors have the option to publish the peer review history of their article (what does this mean?). If published, this will include your full peer review and any attached files.

Reviewer #1: **Yes: **Rafik Margaryan

Reviewer #2: No

---

## [Author Response · Author response to Decision Letter 0]

21 Mar 2023

Comments to the Author

1. Is the manuscript technically sound, and do the data support the conclusions?

Reviewer #1: No

Reviewer #2: Yes

Response: Thank you for pointing this out. 

Our conclusion consists of 

1. Postoperative ischemic injury (shown by the cardiac enzyme results); OPCAB implementation resulted in a lower occurrence of postoperative ischemic injury than ONCAB and ONBHCAB. 

2. Postoperative hemodynamic function (shown by hemodynamic function results); No differences in postoperative hemodynamic function in all three techniques were observed. 

3. Long-term outcomes (shown by the Survival curve and Free from MACE curve in the results); OPCAB respectively were preferable techniques beneficial for long-term outcomes.

That was all data drawn to support the conclusion.

Comment from reviewer 1

Reviewer #1: Congratulations for the arguments that authors have selected.

Comment 1: Materials and Method, Reasoning behind the exclusion from off-pump to on-pump surgery?

Response: Thank you for pointing this out. 

1. Operative techniques were converted intraoperatively from OPCAB to on-pump CABG:

If the procedure was converted to ONBHCAB or ONCAB, it was an unplanned CABG procedure from the beginning. In other cases, the operation may have started as OPCAB but later, nearing the end of the procedure, the patient had to be converted, resulting in adverse outcomes from both OPCAB and on-CPB CABG.

2. Preoperative acute coronary syndrome, acute myocardial infarction:

Since this study requires the collection of cardiac enzymes to compare between groups, it may be difficult to interpret the rise in CK-MB and troponin levels if patients have an acute coronary syndrome or acute MI, as the increase in enzyme levels may not be solely attributed to postoperative values.

3. Preoperative shock:

 The same reason as the acute MI, because this study needs to collect the hemodynamic data preoperative and postoperative to compare between groups, if patients have an unstable preoperative hemodynamic function, the postoperative value may be difficult to interpret.

4. Emergency CABG 

 In emergency cases, the PA catheter is usually not monitored, which means that hemodynamic values measured by the CCO monitor may not be collected. Additionally, emergency CABG surgeries may be complicated by severe acute MI or mechanical complications of MI, which often result in poor hemodynamic function."

Comment 2: What kind of propensity matching algorithm did you use? After matching form smallest group (517 OPCAB) 443 remained.

Response: Thank you for pointing this out. 

A propensity score of multiple arms (3 groups), or the predicted probability of receiving ONCAB, OPCAB, and ONBHCAB, was calculated from the multinomial logistic regression model. (the picture for easier understanding was in the response to the reviewer.)

Step 1. we used multinomial logistic regression to create propensity scores for 3 groups. The variables included in the model for propensity score were described in the statistical analysis issue.

Then we got the propensity score of three groups.

Step 2. Make the pattern that creates from the propensity score in each patient.

After finishing we get patterns like 513, 432, 423, 323 …..

Step 3. Match all three groups with the same pattern (in the same way as conventional propensity score matching, but use a pattern instead of score)

Example 

• Pattern 125 had only one patient in group1, so this one patient from group 1 was removed from the study. (total 1 patient before matching, 0 patient after matching)

• Pattern 224 had two patients in group0, and four patients in group1&2 (a total of 10 patients before matching), after matching group 1&2 reduce to two patients each. (total 6 patients after matching)

The reference for this method of propensity score matching was in the statistical analysis issue.

Comment 3: Results, the quantity of grafts in groups was different?

Response: Thank you for pointing this out. 

We believe that the quantity of coronary anastomoses, as shown in Table 2, is a more important factor for long-term results than the number of graft conduits used. Therefore, we did not provide detailed information on the number of grafts used. However, in our CABG strategy, we typically use one graft conduit for each coronary main branch. When a connection to a sub-branch is needed, we usually connect the sequential graft conduit in the same main branch system. For example, we use LIMA to connect to the LAD and diagonal branch, radial artery to the OM branch and distal LCX, and SVG to the PD and/or PL."

Comment 4: present P-value for survival differences.

Response: Thank you for pointing this out. 

We generated long-term survival curves and estimated freedom from MACE for patients who underwent three different types of CABG using flexible parametric survival regression (STPM2) with time-varying coefficients (TVC). As a result, the survival difference cannot be determined using the log-rank test. However, the hazard ratios (with p-value and 95%CI) for cardiac-related mortality and MACE were calculated using the same regression method. Additionally, in the supporting information (S2 Fig), we provided a crude analysis of Kaplan-Meier survival and freedom from MACE between the three groups, and we added the p-value for survival differences in the figures.

Comment 5: Coronary anastomosis is also taking into account sequential grafting or only graft numbers?

Response: Thank you for pointing this out. 

Coronary anastomosis includes the counting from both sequential grafting (side-to-side anastomosis) and end-to-side anastomosis, and it was not account for the number of graft conduits. 

Comment 6: Very interesting results. Re-fine a little bit your propensity matching technique, usually it is well defined for two groups. In the tables the p-values are post hoc corrected? If not, do so to have a realistic number of p-values.

Response: Thank you for pointing this out. 

The p-values were post hoc analyses because the analysis of variance (ANOVA) with Bonferroni was applied for comparisons between groups.

Comment from reviewer 2

Reviewer #2: i read with great interest the MS by Phothikun et al.

the authors concluded OPCAB implementation resulted in a lower occurrence of postoperative ischemic injury than ONCAB and ONBHCAB. No differences in respectively were preferable techniques beneficial for long-term outcomes.

the MS is technically sound but i have several comments

Comment 1. the authors conducted PS-matching however the process of variables selection is not well defined; i agree n of cases per surgeon should remain adjustment variable but other essential variables (like in Table 1) other than ACEi inhibitors intake should be considered;

Response: Thank you for pointing this out. 

We described in the statistical analysis issue that “The variables included in the model for propensity score were age, sex, New York Heart Association Functional Class, Canadian Cardiovascular Society Classification, the use of angiotensin-converting enzyme inhibitor drugs/angiotensin receptor blockers drugs, the use of pre-operative Aspirin, the pre-operative use of Clopidogrel, end-stage renal disease, coronary artery disease type, and left main coronary disease”. All of these variables were the statistically significant difference when compare between the three groups.

Comment 2. Authors adjust long-term survival with completeness of revascularization which is correct; why not to PS-match the patients with the n. of anastomoses and/or CR

Response: Thank you for pointing this out. 

We used propensity scores based on the principle that the score refers to the chance of patients being selected into each of the three study groups. Therefore, the variable included in the model should be the variable from the preoperative data or baseline characteristics of the patients. We also think other variables based on perioperative or post-operative data should be used for adjustment in the statistical calculations instead of being included in the propensity score.

Comment 3. authors mention converted patients were excluded from the study, but how many there were? it seems from the total numbers, OPCAB is not preferred technique (25% of total); are there any factors other than surgeons' preference to drive patients towards one or another approach?

Response: Thank you for pointing this out. 

"62 OPCAB patients (11.9%) were excluded from the study because they were converted to on-CPB CABG. There may have been several factors involved, such as the severity of aortic calcification, stage of kidney disease, LVEF, The severity of coronary stenosis, number of planned anastomosis sites, area of ischemia, and availability of graft conduits. Ultimately, the technique chosen depended on each surgeon's decision."

Comment 4. OPCAB is further performed predominantly by one surgeon, with higher rates of complete revascularization in this group, contra-intuitively, it poses a limitation; the results of improved survival and reduced ischemia can, therefore, be translatable to broader population of patients only provided that surgeons performing OPCAB have experience in OPCAB and can offer complete revascularization.

Response: Agree, Thank you for pointing this out. 

As this was a retrospective study, the surgeon's experience and skills could have potentially introduced bias in comparative studies of CABG. To address this, we included the surgeon variable as an adjustment variable in our statistical calculations, taking into account differences in the number of CABG techniques used by each surgeon. We used the number of cases performed by each surgeon as a proxy for their experience, which was included in the analysis.

With regard to the completeness of revascularization, our center does not adhere to a strict definition. However, we have a policy of revascularizing all graftable targets in every territory for every case, if feasible, irrespective of techniques. In our study, the mean number of coronary anastomoses performed was 4, which is considered adequate compared to previous studies.

Comment 5. language editing is required

Response: Thank you for pointing this out. 

In addition to the comments, all spelling and grammatical errors have been corrected again by the staff of the Chiang Mai university English language team (CELT) for language editing assistance.

We look forward to hearing from you in due time regarding our submission and to responding to any further questions and comments you may have. 

Sincerely, yours

---

## [Decision Letter · Decision Letter 1]

25 Apr 2023

PONE-D-23-02554R1The Outcomes of Three Different Techniques of Coronary Artery Bypass Grafting: On-pump Arrested Heart, On-pump Beating Heart, and Off-pump.PLOS ONE

Dear Dr. Tepsuwan,

Thank you for submitting your manuscript to PLOS ONE. After careful consideration, we feel that it has merit but does not fully meet PLOS ONE’s publication criteria as it currently stands. Therefore, we invite you to submit a revised version of the manuscript that addresses the points raised during the review process.

ACADEMIC EDITOR:The authors should add a new paragraph of discussion regarding perspectives on new early biomarkers of outcome in CABG patients. Recent study has demonstrated that plasma exosome levels increase at 3h and 72h after aortic clamping in older patients undergoing first-time on-pump coronary artery bypass graft and play an antiapoptotic role. In particular, troponin levels did not increase at 72h early after CABG (please see Geroscience 2021 Apr;43(2):773-789). These findings suggest the role of plasma exosomes as perioperative biomarker of tolerance against the perioperative ischemic insult of the heart in CABG patients, as demonstrated by other study (please see PLoS One. 2016 Apr 29;11(4):e0154274.; Acta Anaesthesiol Scand. 2019 Apr;63(4):483-492.) . Please add and discuss the abovementioned studies.

We look forward to receiving your revised manuscript.

Kind regards,

Vincenzo Lionetti, M.D., PhD

Academic Editor

PLOS ONE

Journal Requirements:

Reviewers' comments:

Reviewer's Responses to Questions

**Comments to the Author**

1. If the authors have adequately addressed your comments raised in a previous round of review and you feel that this manuscript is now acceptable for publication, you may indicate that here to bypass the “Comments to the Author” section, enter your conflict of interest statement in the “Confidential to Editor” section, and submit your "Accept" recommendation.

Reviewer #1: All comments have been addressed

2. Is the manuscript technically sound, and do the data support the conclusions?

Reviewer #1: Yes

3. Has the statistical analysis been performed appropriately and rigorously? 

Reviewer #1: Yes

4. Have the authors made all data underlying the findings in their manuscript fully available?

Reviewer #1: No

5. Is the manuscript presented in an intelligible fashion and written in standard English?

Reviewer #1: Yes

6. Review Comments to the Author

Reviewer #1: Comments are answered properly, please do make available de-identified data (see plos one data policy).

7. PLOS authors have the option to publish the peer review history of their article (what does this mean?). If published, this will include your full peer review and any attached files.

Reviewer #1: **Yes: **Rafik Margaryan

---

## [Author Response · Author response to Decision Letter 1]

2 May 2023

Dear Vincenzo Lionetti, Acedemic Editor

Thank you for giving us the opportunity to re-submit a second revised draft of our manuscript titled “Outcomes of three different techniques of coronary artery bypass grafting: on-pump arrested heart, on-pump beating heart, and off-pump” to PLOS ONE. We appreciate the time and effort you and the reviewers have dedicated to providing valuable feedback on our manuscript. We are grateful to the reviewers for their insightful comments on our paper. We have been able to incorporate changes to reflect most of the suggestions provided by the reviewers. 

Here is a point-by-point response to the academic editor’s and the reviewer’s comments and concerns. 

ACADEMIC EDITOR: The authors should add a new paragraph of discussion regarding perspectives on new early biomarkers of outcome in CABG patients. Recent study has demonstrated that plasma exosome levels increase at 3h and 72h after aortic clamping in older patients undergoing first-time on-pump coronary artery bypass graft and play an antiapoptotic role. In particular, troponin levels did not increase at 72h early after CABG (please see Geroscience 2021 Apr;43(2):773-789). These findings suggest the role of plasma exosomes as perioperative biomarker of tolerance against the perioperative ischemic insult of the heart in CABG patients, as demonstrated by other study (please see PLoS One. 2016 Apr 29;11(4):e0154274.; Acta Anaesthesiol Scand. 2019 Apr;63(4):483-492.) . Please add and discuss the abovementioned studies.

Response: Thank you for pointing this out. 

We add a new paragraph of discussion regarding perspectives on “exosome” outcomes in CABG as you recommend. The reference was also added as the same recommendation.

The passage below was the added new passage to the discussion issue.

 “In recent times, new biomarkers have emerged that can be compared to cardiac troponin, the current 'gold standard' surrogate biomarker of myocardial damage. Plasma exosomes are small extracellular vesicles that are released into the bloodstream and contain various molecular cargoes that play a role in intercellular communication. During cardiac surgery, ischemia-reperfusion injury can lead to tissue damage and inflammation, resulting in exosome release from injured or stressed cells [16]. Emanueli et al. studied exosomes containing cardiac microRNAs and found that plasma concentrations of exosomes and their cargo of cardiac microRNAs increased in patients undergoing CABG and correlated with cardiac troponin levels [17]. Carrozzo et al. studied between plasma exosomes and serum cardiac troponin I levels in older (on-CPB) CABG patients. After aortic de-clamping, exosome levels significantly increased at both 3 hours and 72 hours, while troponin I level peaked at 3 hours and then gradually decreased [18]. Moreover, Frey et al. discovered that exosome release was related to ischemic/reperfusion injury and associated with myocardial tissue protection [19]. Therefore, due to their persistence and specific biomarkers for cardiac injury, exosomes have the potential to be a valuable new standard tool in the detection and monitoring of myocardial injury in the near future.”

Comments to the Author

Comment from Reviewer 1

Reviewer #1: 

Comment 4. Have the authors made all data underlying the findings in their manuscript fully available?

Reviewer #1: No

Response: Thank you for pointing this out. 

The new shared data set (data set ver.2) was the data before we had done the propensity matching. We shared the supplyment1 Data “The minimal data set used for analyzed” with the data availability statement: All relevant data are within the paper and its Supporting Information files. Moreover, the new minimal data set can access via; https://doi.org/10.6084/m9.figshare.22723688.v1

Comment 6. Review Comments to the Author

Reviewer #1: Comments are answered properly, please do make available de-identified data (see plos one data policy).

Response: Thank you for pointing this out. 

We made available de-identified data in the new shared data set (data set ver.2).

We look forward to hearing from you in due time regarding our re-submission and to respond to any further questions and comments you may have. 

Sincerely, yours

Amarit Phothikun, MD

Cardiovascular and Thoracic Surgery Unit,

Department of Surgery Faculty of Medicine,

Chiang Mai University, Chiang Mai, Thailand

110 Intravaroros Road, Sriphum, Mueng,

Chiang Mai 50200, Thailand 

Email: armadillos176@gmail.com Tel: +66 89 6333 66 1 

Thitipong Tepsuwan, MD

Cardiovascular and Thoracic Surgery Unit,

Department of Surgery Faculty of Medicine,

Chiang Mai University, Chiang Mai, Thailand

110 Intravaroros Road, Sriphum, Mueng,

Chiang Mai 50200, Thailand 

Email: tepsuwanthitipong@gmail.com Tel: +66815956287

---

## [Editor Report · Decision Letter 2]

18 May 2023

The Outcomes of Three Different Techniques of Coronary Artery Bypass Grafting: On-pump Arrested Heart, On-pump Beating Heart, and Off-pump.

PONE-D-23-02554R2

Dear Dr. Tepsuwan,

We’re pleased to inform you that your manuscript has been judged scientifically suitable for publication and will be formally accepted for publication once it meets all outstanding technical requirements.

Kind regards,

Vincenzo Lionetti, M.D., PhD

Academic Editor

PLOS ONE
---

## [Editor Report · Acceptance letter]

22 May 2023

PONE-D-23-02554R2 

The outcomes of three different techniques
of coronary artery bypass grafting: on-pump arrested heart,
 on-pump beating heart, and off-pump 

Dear Dr. Tepsuwan:

I'm pleased to inform you that your manuscript has been deemed suitable for publication in PLOS ONE. Congratulations! Your manuscript is now with our production department. 

Kind regards, 

on behalf of

Prof. Vincenzo Lionetti 

Academic Editor

PLOS ONE